# Effect of Carotenoids on Paraoxonase-1 Activity and Gene Expression

**DOI:** 10.3390/nu14142842

**Published:** 2022-07-11

**Authors:** Aneta Otocka-Kmiecik

**Affiliations:** Department of Experimental Physiology, Medical University of Lodz, 6/8 Mazowiecka St., 92-215 Lodz, Poland; aneta.otocka-kmiecik@umed.lodz.pl

**Keywords:** antioxidant, paraoxonase, PON1, carotenoids, astaxanthin, β-carotene, lycopene

## Abstract

Paraoxonase 1 (PON1) is an antioxidant enzyme attached to HDL with an anti-atherogenic potential. It protects LDL and HDL from lipid peroxidation. The enzyme is sensitive to various modulating factors, such as genetic polymorphisms as well as pharmacological, dietary (including carotenoids), and lifestyle interventions. Carotenoids are nutritional pigments with antioxidant activity. The aim of this review was to gather evidence on their effect on the modulation of PON1 activity and gene expression. Carotenoids administered as naturally occurring nutritional mixtures may present a synergistic beneficial effect on PON1 status. The effect of carotenoids on the enzyme depends on age, ethnicity, gender, diet, and PON1 genetic variation. Carotenoids, especially astaxanthin, β-carotene, and lycopene, increase PON1 activity. This effect may be explained by their ability to quench singlet oxygen and scavenge free radicals. β-carotene and lycopene were additionally shown to upregulate PON1 gene expression. The putative mechanisms of such regulation involve PON1 CpG-rich region methylation, Ca(2+)/calmodulin-dependent kinase II (CaMKKII) pathway induction, and upregulation via steroid regulatory element-binding protein-2 (SREBP-2). More detailed and extensive research on the mechanisms of PON1 modulation by carotenoids may lead to the development of new targeted therapies for cardiovascular diseases.

## 1. Introduction

Paraoxonase 1 (PON1) is an enzyme which manifests anti-atherosclerotic properties due to its ability to inhibit low-density lipoprotein (LDL) oxidation [1]. It also exhibits the properties of hydrolyzing organophosphate insecticides and a variety of lactones including homocysteine thiolactone (HCTL) [2]. Therefore, there is an ongoing search for factors, which could modulate its serum activity. One strategy involves the administration of exogenous compounds that could increase endogenous PON1 activity. The second approach is based on the upregulation of gene expression. This could potentially be conducted at different levels: transcriptional and post-transcriptional (in the nucleus) and translational and post-translational (in the cytoplasm). This type of modification is analyzed in a rather limited number of studies, mainly describing modification at the transcriptional, and rarely at post-translational, levels. Very little research has been conducted on the epigenetic regulation of PON1 activity.

At the transcriptional level, the factor specificity protein 1 (Sp1) plays an essential role in PON1 regulation [3]. It can be activated by protein kinase C (PKC) and p44/p42 mitogen-activated protein kinase (MAPK) [4]. P44/p42 MAPK is able to activate another transcriptional factor, sterol regulatory element-binding protein 2 (SREBP 2) [5]. Some polyphenols, such as resveratrol, were found to activate the aryl hydrocarbon receptor (AhR) and stimulate PON1 transcription [6]. Another transcriptional factor, c-Jun, either induces or suppresses the transcription of PON1 depending on its modulating factors working through different signaling pathways [7]. Inhibition of transcription factor kappa-B, which stimulates genes responsible for the production of some inflammatory cytokines, restores PON1 mRNA levels [8].

Post-translational modification of PON1 involves N-linked protein glycosylation in four potential sites (Asn 227 and Asn 270; Asn 253 and Asn 324) [9]. Asn 253 and Asn 324 are most probably the glycosylation sites as they are positioned most superficially on the protein. Some data indicate that glycosylation of the enzyme is not important for its hydrolytic activities but for increasing its solubility and stability.

Among the factors which may affect PON1 activity, carotenoids are listed. They were initially appraised due to their free radical quenching ability [10]. It was soon found that they reduce lipid peroxidation and protect the membrane from oxidative damage. It has been suggested that the antioxidant function of carotenoids may also be mediated through the modulation of PON1 activity and gene expression. While some data on the modulation of endogenous PON1 activity by carotenoids are available, there is a scarcity of research on their influence on PON1 gene expression.

The aim of the review is to compare the effect of carotenoids studied so far. It aims to identify which carotenoids have the strongest effect for consideration in practice when composing diets for people with cardiovascular diseases. In addition, the study objective is to present the current state of knowledge on the molecular mechanism of action of these carotenoids on gene expression and PON1 activity and possibly to indicate the direction of further research in this field. To the best of my knowledge, a review gathering the data and analyzing the influence of carotenoids on PON1 status was lacking. The study will add valuable evidence, which can be useful in atherosclerosis prevention. Understanding the physiological mechanisms which enhance the enzyme activity, affect epigenetic regulation and modify the signaling pathways of PON1 transcriptional factors is crucial for the development of new targeted therapies for cardiovascular diseases.

Exogenous antioxidants are naturally supplied with diets as nutritional mixtures. From a scientific point of view, in this arrangement, it is difficult to single out compounds responsible for specific effects. Due to these concerns, a lot of effort has been put into defining the most relevant chemical compounds responsible for the registered changes in the nutritional mixtures when specific diets are analyzed. In this respect, the most valuable studies show the effects of individual dietary components. A preliminary review of the carotenoid literature shows that there are not many such publications, so in this review, I refer to all available items, including those on the effect of diet with a mixture of carotenoids.

## 2. Properties of Paraoxonase

PON1 (aryldialkylphosphatase, E.C.3.1.8.1) is a calcium ion-dependent enzyme that is able to hydrolyze lactones, thiolactones, aryl esters, and organophosphate derivatives [11]. It is a glycoprotein formed by 354 amino acids with a molecular mass of 43–47 kDa. Its three-dimensional structure is that of a six-bladed β-propeller, of which each blade consists of four strands. Two Ca^2+^ ions are incorporated into the central part of the tunnel of the propeller [9]. One plays a structural and the other a catalytic role. In the active site, the catalytic Ca^2+^ interacts with Asn 168, 224, 270, Asp 269, and Glu 53. The active site of PON1 also contains histidine, tryptophan, phenylalanine, lysine, and aspartate/glutamine [12]. The enzyme has three cysteine residues at positions 42, 284, and 353. Cys42 and Cys353 form a disulfide bridge, while Cys284 remains free. The latter is considered to have an important antioxidant role in protecting LDL against oxidation [1,9]. PON1 is thought to contain polar residues for high-density lipoprotein (HDL) binding. Most of the PON1 in circulation is found attached to HDL but some particles are also associated with very low-density lipoproteins (VLDL) and chylomicrons [13]. PON1 is anchored to HDL by apolipoprotein A-I, which was shown to stabilize the enzyme and stimulate its lactonase activity [14]. A part of the beneficial effects of HDL is attributed to the antioxidant properties mediated by paraoxonase 1 [15].

### 2.1. Paraoxonase Family

PON1 is the best-known member of a three-gene family: PON2, PON3, and PON1, located in this order on the long arm of human chromosome 7 (q21.22) [16]. The paraoxonase family was formed by a duplication of one common ancestral gene. PONs are evolutionarily linked to lactonases, with some overlapping substrates [17]. The name paraoxonase comes from PON1’s ability to hydrolyze paraoxon, a metabolite of the insecticide parathion [18]. It also degrades another artificial substrate, phenylacetate. The name of PON2 and PON3 is a simple derivative of PON1, based on the evolution of the enzymes rather than their activities, as PON2 and PON3 are incapable of hydrolyzing paraoxon. Common features in the whole PON family are the residues maintaining the hydrophobic core of the β-propeller, the two calcium ions in the central tunnel, and the ‘velcro’ closure [9]. The elements forming the catalytic site such as the catalytic calcium and its ligating residues, as well as residues forming hydrogen bonds with calcium ions, and the catalytic histidines are also highly conserved. Cys284, which is proposed to have an important effect of protecting LDL from oxidation, is common to the whole PON family. Thus, the active site and the main catalytic center are maintained among the PON family. Yet, the residues outside the active site are different among subfamilies. This affects substrate specificity. For example, one of the presumed glycosylation sites, Asn253, is found in PON1 but not in PON2 and PON3 [9]. Other residues are most likely specific for each subfamily (such as in the region of positions 20–50), which can affect the nonhydrolytic activities of the PON enzyme and its prevalence in different tissues [16]. PON2 gene is widely expressed in a variety of tissues (brain, liver, kidney, and testis), and it is not released into serum [19]. It is located in the cell membrane with its active site pointing outside of the cell [14]. In contrast, PON1 and PON3 mRNA expression and synthesis are limited mainly to the liver and then excreted to the blood. PON1 and very low levels of PON3 are found in serum bound to HDL by the hydrophobic N-terminal leader sequence [20]. All PON enzymes have an antioxidant capacity and are known to decrease the risk of atherosclerosis development [21,22]. Despite having that effect, the physiological substrates of PONs remain unknown [23]. Some of the proposed substrates include oxidized 1-palmitoyl-2-arachidonoyl-sn-glycerol-3-phosphorylcholine (Ox-PAPCP) [24], cholesterol linoleate hydroperoxide [25], oxidized linoleic acid [26], platelet-activating factor (PAF) [27], and HCTL [2]. It is generally accepted that paraoxonase has a wide range of biological substrate specificity.

In human serum, PON1 activity predominates. It travels in the blood attached to the HDL particle and exerts its antioxidant effects. Its ability to inhibit lipid peroxidation, among other activities, resulted in numerous studies on the association of PON1 with atherosclerosis and cardiovascular disease. In this review, we focus on PON1, as current research shows that this paraoxonase family member has the most significant impact on human pathophysiology. We attempt to assemble evidence on the ability of carotenoids to modulate PON1 activity.

### 2.2. Anti-Atherosclerotic Effect of PON1

Many studies support the hypothesis that low PON1 activity is associated with increased oxidative stress, risk of atherosclerosis, and cardiovascular disease. Some prospective studies such as the Caerphilly Prospective Study, and a study conducted by Bhattacharyya show that not only HDL itself but also PON1 is an independent risk factor for coronary artery disease [28,29]. PON1 activity was found to be the lowest in patients after acute myocardial infarction, higher in stable coronary disease, and the highest in controls [30]. Several mechanisms by which PON1 can delay and reverse atherosclerosis progression were detected in mice overexpressing PON1 [31] and mice lacking PON1 [32], as well as in vitro studies on lipoproteins or macrophage cell lines. The anti-atherogenic mechanisms of PON1 are summarized in Figure 1.

There is a body of evidence proving that PON1 protects LDL and HDL and cell membranes from oxidative modification and inhibits the progression of atherosclerosis [1,33]. It can hydrolyze specific oxidized lipids such as phosphatidylcholine core aldehydes and oxidized eicosanoids and docosanoids in LDL and the cell membranes [25]. The same hydrolytic activity of PON1 was observed in atherosclerotic lesions [34] and arterial wall cells [35]. The inhibition of HDL oxidation by PON1 was shown to preserve the anti-atherogenic properties of HDL. HDL isolated from mice was able to prevent LDL oxidation, while HDL from PON1 knockout mice lacked that ability [32]. Furthermore, HDL isolated from mice overexpressing PON1 was even more efficient in the inhibition of LDL oxidation than HDL from wild-type mice [31]. These observations are supported by studies on human subjects. Low activity of PON1 and high levels of lipid peroxides were measured in patients with metabolic syndrome [36].

PON1 protects macrophages from oxidative stress and stimulates cholesterol efflux. PON1 can be delivered to macrophages during HDL interaction with the surface of macrophages, which protects them from oxidative stress [37]. The enzyme inhibits cholesterol biosynthesis and accumulation in macrophages [38]. It induces the production of lysophosphatidylcholine (LPC) from phosphatidylcholine, which in turn inhibits the formation of superoxide anions and reduces cell-mediated LDL oxidation [39]. These changes reduce foam cell formation in macrophages.

It was also found that PON1 has the ability to hydrolyze HCTL, which is a highly reactive metabolite of homocysteine (a sulfur non-protein amino acid). It is formed in cells by methionyl-tRNA synthetase in a two-step error-editing reaction, which prevents the incorporation of homocysteine into proteins. Increased levels of homocysteine result in the elevation of HCLT. It is considered a risk factor for cardiovascular disease [2]. HCTL is responsible for the N-homocysteinylation of proteins and lipoproteins, including LDL and HDL. This process damages their structure and impairs their function [2]. PON1 is able to detoxify this reactive metabolite.

In addition to inhibiting early plaque formation, PON1 was found to lower oxidative stress and improve vasomotor function during established atherosclerosis [40].

### 2.3. PON1 Polymorphism

The activity of PON1 depends on its genetic polymorphisms, which are responsible for its almost 40-fold variations. The difference in PON1 protein levels vary up to 13–15-fold [41]. Over 200 single-nucleotide polymorphisms (SNPs) of PON1 in different regions of the gene were detected [9]. The highest impact of these polymorphisms on the activity and protein level of the enzyme is attributed to SNPs of the coding region at positions 192, 55, and the −108 promoter region as presented in Table 1. These polymorphisms result in different isoforms of the enzyme and a different hydrolytic activity of the enzyme isoforms towards various substrates. For example, in case of glutamine (Q)/arginine (R) substitution at codon 192, the Q isoform has a higher hydrolytic activity towards diazoxon (diazoxonase activity) and the R isoform towards paraoxon (paraoxonase activity) [42]. The hydrolysis rate of another substrate, phenylacetate (arylesterase activity), is the same for both allozymes [43]. More importantly, PON1 isoforms vary in retarding the oxidation of LDL. In particular, the Q allele is far more efficient than the R allele in protecting LDL from oxidation [44]. The leucine (L)/methionine (M) substitution at position 55 results in different plasma PON1 protein levels. The M allele is associated with a low PON1 plasma protein level. In addition, the T/C substitution at the −108 promoter region appears to influence the plasma PON1 protein levels. The low plasma PON1 protein level in PON1M55 may result from linkage disequilibrium with the C-108T allele. In case of the C-108C allele, PON1 levels are twice as high as in case of the C-108T allele [45].

Many studies prove an association between PON1gene polymorphisms and cardiovascular disease [55,56]; yet in others, this relation is not found [57]. This lack of association in some observations can be explained, at least in part, by the susceptibility of PON1 gene expression and activity to different modulating factors. Therefore, a consensus was formed that PON1 concentration and activity are higher predictors of cardiovascular disease than the PON1 genotype alone [43]. Furthermore, considering that promoter polymorphisms are associated with the early onset of coronary artery disease [58], a search had begun to identify interventions, which could enhance the expression and activity of the enzyme. Studies show that PON1 can be influenced by pharmacological treatment, environment, lifestyle, and diet.

### 2.4. The Influence of Environmental Factors on PON1 Activity and Concentration

In studies on humans, hypolipidemic pharmacological drugs such as statins and fibrates were shown to modulate PON1 activity. Simvastatin and atorvastatin [48,59,60] cause an increase in serum PON1 activity. However, a lack of influence of statins on the enzyme activity was also found in another study [61]. Fibrates (gemfibrozil, fenofibrate, ciprofibrate) induced PON1 activity in serum and isolated HDL [62,63,64]. Cigarette smoking was associated with a decrease in PON1 activity [65].

In search of more natural ways of influencing PON1 activity, research began to focus on lifestyle interventions, which could affect the enzyme such as physical activity. Many but not all studies show that a single exercise leads to an increase in PON1 concentration and activity in the plasma of young men [66,67,68,69]. Yet, this increase in activity is not stable. A decrease or at least a return to basal levels within two hours after exercise was observed. Training, however, consolidates the changes in PON1 status. The exercise conducted regularly for 8–10 years improved PON1 activity even at rest [70]. Subjects who undertook regular physical activity had higher PON1 activity than sedentary subjects [69,71]. This implies that some mechanisms evoked by physical activity stimulate PON1. Free radicals released during physical exercise may upregulate antioxidant enzyme expression [72].

It has been suggested that the consumption of red wine or flavonoid-containing drinks increases serum PON1 activity [73]. Yet, a high dosage of red wine polyphenols decreased hepatic PON1 activity in mice, even though a lower dosage had a beneficial effect on the enzyme activity [74]. The consumption of a moderate amount of alcohol (13–39 g/day) caused an increase in PON1 activity. On the contrary, heavy alcohol drinking had a detrimental effect on the enzyme [75].

Moreover, environmental factors such as diet and nutrition can affect the regulation of PON1 at an epigenetic level, causing changes at specific loci, which can modify corresponding phenotypes. Studies concerning this topic are scarce. They have been reviewed by Mahrooz et al. [76]. An increase or decrease in DNA methylation can result in gene silencing or overexpression, respectively. An inverse association between methylation levels of PON1 promoter region CpG sites and ARE in adults with metabolic syndrome was described in a six-month energy-restricted dietary weight-loss intervention [77]. Additionally, this study showed that dietary antioxidants might enhance the ARE activity by lowering the PON1 gene methylation. In another study, an association between methylation at two PON1 promoter CpG sites with body weight and waist circumference was reported, which proves that PON1 DNA methylation may influence obesity risk. The microRNAs (miRNAs) are able to inhibit the expression of genes, i.e., PON1 by binding to 3′-UTR of the coding region of target mRNAs. It was observed that miR-616 negatively regulated the expression of the PON1 gene and protein level. Moreover, miR-486 was found to correlate inversely with PON1 activity. The intense continuous exercise reduced circulating miR-486 [78]. This could potentially explain the rise of PON1 observed at the bout of exercise [68]. Epigenetic regulation of PON1 is a very promising subject, which calls for further research.

### 2.5. The Influence of Various Components of Diet on PON1 Activity and Gene Expression

The effect of diet on many parameters of health maintenance has been a subject of scientific interest for many years and it is still a current topic. As the methods of research in medicine and biology are evolving, we are able to investigate more mechanisms on the cellular and subcellular levels.

The diet has been shown to affect PON1 activity. In a systematic review, Lou Bonafonte et al. introduced evidence that the Mediterranean diet exerts a protective effect on the enzyme [79]. Extra virgin olive oil was found to be particularly effective in increasing PON1 activity. In vivo, olive oil consumption by mice increased PON1 activity in parallel to HDL-mediated macrophage cholesterol efflux. Similar effects of olive oil consumption were observed in healthy humans [80]. Oleic acid was associated with increased HDL cholesterol levels and PON1 activity, especially in patients carrying the R allele. Furthermore, in a study on mice, the consumption of a diet containing high amounts of olive oil phenolics increased hepatic PON1 mRNA and protein expression [81]. In addition, squalene dissolved in virgin olive oil promoted increases in PON1 level with a concomitant rise in HDL and decreases in reactive oxygen species in lipoproteins and a fall in plasma malondialdehyde level [82].

Some compounds being part of fruit and vegetables, such as vitamin C and E, were also found to modify PON1 activity. Vitamin C was shown to preserve the cardio-protective activity of PON1. Yet, this was not accompanied by restoring HDL ability to prevent atherogenic modification of LDL [83]. Vitamin C attenuated the inhibitory effect of hypochlorite on PON1 activity in vitro [84]. Vitamin E was found to improve PON1 activity in patients with type 2 diabetes [85]. Yet, the effect of vitamin supplementation was not always shown to be beneficial. A reduction in PON1 activity after a high intake of fruits and vegetables, which was expected to increase vitamin C and E, was also observed [86]. In general, vitamin C and E should be supplemented in small amounts. If excess levels of these antioxidants are achieved, they can have a detrimental effect and result in a rise in oxidation [87].

In addition, other groups of antioxidants acquired from fruit and vegetables, i.e., phenolic compounds and carotenoids, were found to be particularly effective in increasing PON1 activity [88,89,90]. These nutritional antioxidants may induce PON1 activity through effects on gene expression, increase in PON1 gene activation, prevention of PON1 inactivation, and binding of PON1 to HDL and increasing its stability [91]. Pomegranate juice consumption, as a source of phenolic antioxidants, enhanced PON1 activity [92,93,94]. Other fruits were not examined as thoroughly, yet raspberry juice and apple juice were shown to increase PON1 activity in animal studies [95,96]. Not all fruit juices were shown to have this beneficial effect. In human studies, orange and blackcurrant juices did not influence the enzyme activity [97]. It is quite well documented that polyphenols are promising antioxidants, which may modulate PON1 activity and gene expression [98]. Quercetin increased enhanced PON1 hepatic expression and PON1 activity in the liver and serum [99]. Aqueous extracts of yerba mate, which has a high concentration of polyphenols, can contribute to the improvement of PON1 levels in individuals affected by overweight or obesity and dyslipidemia [100]. The positive effect of a diet rich in fruit and vegetables is also associated with another group of antioxidant components, i.e., carotenoids.

## 3. Properties of Carotenoids

One of the groups of nutritional agents mentioned above, which is known for its protective properties against oxidative damage is carotenoids. They are represented by a wide variety of compounds. Carotenoids are natural pigments with lipophilic properties. They are synthesized by plants de novo, and so they are administered with a diet, mostly in fruit and vegetables. These compounds are also distributed in foods of animal origin, as compounds accumulated from plants, sometimes in a slightly changed form [101]. Carotenoids are transported in human blood attached to plasma lipoproteins, mainly LDL particles.

The reason why they may serve as antioxidants is in their structure. All carotenoids have a polyene backbone consisting of a series of conjugated C=C bonds. This offers an opportunity for many carotenoids to interact with free radicals and singlet oxygen and fight oxidative stress. The reactivity of carotenoids varies depending on modifications to this polyene backbone, such as the number of conjugated double bonds and the addition of oxygen functional groups [10,71]. Here, in this review, we focus on carotenoids, which were found to have a protective effect on the cardiovascular system, i.e., α-carotene, β-carotene, zeaxanthin, astaxanthin, lutein, β-cryptoxanthin, and lycopene. Carotenes such as α-carotene, β-carotene, and lycopene are transported mainly in the inner part of LDL. β-cryptoxanthin, lutein, and zeaxanthin are classified as xanthophylls, and they are transported on the outer surface area of LDL and HDL [102,103,104]. This specific localization of carotenoids together with their antioxidant properties allows them to help prevent atherosclerosis. Major dietary carotenoids are presented in Table 2.

A study on nearly 400 subjects showed a reduction in the risk of atherosclerosis due to carotenoids administration [105]. Another study indicates that some carotenoids, i.e., β-cryptoxanthin and lutein reduce the risk of acute myocardial infarction [106]. A study on coronary mortality in 16 countries showed that a diet low in food containing folate and lutein/zeaxanthin might be an important factor contributing to a higher coronary risk observed in Central and Eastern Europe [107]. Early atherosclerosis patients had lower serum concentrations of lutein and zeaxanthin than healthy subjects [108]. In a cohort study conducted on 3116 Japanese patients, higher levels of serum carotenoid were associated with lower risks of all-cause, cancer, and cardiovascular disease mortality in Japanese patients [109]. Diabetic patients with higher serum carotenoid concentrations had fewer vascular complications [110]. However, the mode of action of carotenoids in vascular endothelial cells is still not fully understood. It has been suggested that these compounds may activate an HDL-like protective mechanism in endothelial cells. Additionally, carotenoids protect human lymphocytes from oxidative damage and decrease the risks of some chronic diseases and degenerative disorders including some cancer types [111]. There is some evidence that some of the beneficial effects of carotenoids on the human body are achieved through increasing paraoxonase activity.

## 4. The Influence of Carotenoids on PON1 Activity and Gene Expression

In this review, we focus on the research on the influence of carotenoids on PON and ARE activities of PON1 and PON1 gene expression. Most reviewed literature on the influence of carotenoids on PON1 focuses on astaxanthin, β-carotene, and lycopene. These carotenoids are easily accessible and highly available in the diet. Previous studies show that they have pronounced antioxidant and atheroprotective effects. Astaxanthin effectively scavenges free radicals, thereby protecting fatty acids and biological membranes from oxidative damage [112]. Lycopene has a high antioxidant capacity as the β cycle in its structure is opened [113]. β-carotene is inversely associated with atherosclerosis in various vascular territories [105]. Table 3 summarizes the results of animal studies, while Table 4 provides an overview of clinical studies on the effect of carotenoids on PON1. Further description of the studies is supplied in the main text.

### 4.1. The Influence of Astaxanthin on PON1 Activity

Astaxanthin is a carotenoid pigment synthesized by plants and some bacteria, algae, and fungi and distributed in some fish such as salmon and trout as well as crustaceans [103,130]. It is an antioxidant, which serves as a free radical scavenger. It protects fatty acids and cell membranes from oxidative damage [131]. It was shown to reduce lipid peroxidation while preserving the membrane structure [132].

#### 4.1.1. The Influence of Astaxanthin on PON1 Activity in Animal Studies

PON1 activity was inhibited in parallel to LDL oxidation in the serum of hypercholesterolemic rabbits. Supplementation of diet with astaxanthin restored PON1 activity [112]. This effect may be explained by a mechanism that was introduced in a dynamic model based on the Atlantic salmon system, where the antioxidant was transported in the bloodstream from LDL and VLDL to HDL [130]. Astaxanthin may have an attachment site near PON1 in the HDL particle [112]. Due to this location on HDL, it can exert its protective effect on the enzyme. Even though PON1 activity was preserved by the carotenoid, no protection of LDL oxidation was registered. This observation may have been related to a lower PON1/HDL ratio in hypercholesterolemic rabbits in comparison to a control group. It may also be due to other factors influencing LDL oxidation that are not under the control of astaxanthin [112]. Kukurt et al. described a protective effect of astaxanthin in a study on 3-nitropropionic-acid-induced ovarian damage in rats [114]. As the destruction of ovaries can be explained by an oxidative mechanism, treatment with antioxidants may decrease the negative changes in structure. Indeed, administration of astaxanthin resulted in an improvement of histopathological ovarian damage. These beneficial changes were accompanied by a restoration of PON1 activity with a concomitant rise in total antioxidant capacity, whole blood reduced glutathione, and HDL, as well as a reduction in total oxidant capacity and oxidative stress index. These changes speak for the antioxidant properties of astaxanthin.

#### 4.1.2. The Influence of Astaxanthin on PON1 Activity in Clinical Studies

In human studies, 90 days of carotenoid supplementation in young elite soccer players during their training program resulted in an increase in PON1 activity and improvement of the activity towards paraoxon and diazoxon with a concomitant rise in total sulphhydryl group content [119]. These changes were not observed in a group receiving a placebo. It was previously observed that exposure of PON1 to hydroxyl radicals and superoxide anions caused a fall in PON1 activity and the number of PON1-free thiol groups [133]. The authors suggest that astaxanthin supplementation might increase total sulphhydryl group content. PON1 has a free cysteine residue (Cys284), which was shown to be important for the enzyme’s activity [134]. The rise of PON1 activity may be caused by the protection of free thiol groups in the active center of the enzyme against oxidative damage. In disease states that lead to a reduction in PON1 activity, astaxanthin was shown to restore PON1 activity. In addition, astaxanthin supplementation may be useful in augmented antioxidant demand, such as during the training season of soccer players. It may deliver additional antioxidant protection and increase PON1 activity.

At present, no data exist on the influence of astaxanthin on PON1 gene expression.

### 4.2. The Influence of β-Carotene on PON1 Activity and Gene Expression

Another carotenoid, β-carotene, is found in palm fruits, squash cultivars, green vegetables, carrots, orange-fleshed sweet potato, cantaloupe, mango, and apricot [103].

#### The Influence of β-Carotene on PON1 Activity and Gene Expression in In Vitro Studies

β-carotene strongly induced gene expression of PON-1 in cultured human endothelial cell lines [135]. A key role in the formation of initial arteriosclerotic lesions is played by an inflammatory interleukin-1 beta (IL-1β), which induces endothelial dysfunction [136]. It was found that the promoter activities of PON1 were downregulated by IL-1β in HepG2 cells [137]. IL-1β decreased the activity of PON-1, which may have negatively impacted the protection from oxidative stress in endothelial cells. To prevent the development of atherosclerosis, it is important to inhibit IL-1β-mediated endothelial alterations and upregulate protective mechanisms [138,139]. The addition of β-carotene to confluent endothelial cells treated with IL-1β was able to reverse the effects of IL-1β on the gene expression of PON-1 via Ca(2+)/calmodulin-dependent kinase II (CaMKKII) pathway induction. It led to an increase in PON-1 protein expression [135]. Enhanced adherence of monocytic U937 cells to human aortic endothelial cells was observed after treatment with IL-1β [140]. The addition of β-carotene, lutein, and lycopene led to a reduction in the adhesion. In conclusion, β-carotene may induce PON1 activity and gene expression and reduce endothelial cell dysfunction caused by inflammatory cytokines such as IL-1β through a mechanism similar to HDL and may reinforce the effects of HDL.

### 4.3. The Influence of Lycopene on PON1 Activity and Gene Expression

Lycopene is a carotenoid present in tomato, pitanga, pink-fleshed guava, red-fleshed papaya, and watermelon. The richest source of lycopene is the Asian gac fruit and the Spanish sarsaparilla [103]. It is considered to possess the most potent antioxidant activity of all carotenoids in accordance to the following ranking: lycopene > α-carotene > β-cryptoxanthin > zeaxanthin = β-carotene > lutein [141]. Its antioxidant effects are summarized in Figure 2.

Lycopene has especially high free radical scavenging properties which can be explained by a high number of conjugated double bonds with a high singlet oxygen quenching ability [142]. Experimental evidence shows that lycopene can quench singlet oxygen (102), scavenge free nitrogen dioxide (NO•2), thiyl (RS•), and sulfonyl (RSO•2) radicals [147]. Due to its ability to catch free radicals and decrease the damage caused by oxidative stress in lipids, lipoproteins, proteins, and DNA, it was suggested to prevent atherogenesis and carcinogenesis [143]. Lycopene lacks hydrophilic substituents, and therefore, it is very hydrophobic. It has been strongly associated with the ability to decrease LDL oxidation and overall lipid peroxidation [150]. High consumption of tomato products resulted in a decrease in LDL cholesterol level and an increase in LDL resistance to oxidation in healthy normocholesterolemic adults [144]. These atheroprotective changes correlated with an increase in lycopene, α-carotene, and β-carotene levels measured in serum.

#### 4.3.1. The Influence of Lycopene on PON1 Activity and Gene Expression in Animal Studies

Hypercholesterolemia induces oxidative stress. In a study on hyperlipidemic rats, lycopene supplementation was shown to restore plasma antioxidant levels measured as the ferric-reducing activity of plasma (FRAP), which was accompanied by a rise in PON1 arylesterase activity [115]. The PON1 enzyme is thought to be partly responsible for the rise in FRAP. The observed improvement in PON1 activity may have been achieved by an upregulation of PON1 gene expression. PON1 expression was previously seen to be upregulated by transcription factors such as steroid regulatory element-binding protein-2 (SREBP-2), which binds to the promoter region of PON1 [48]. Apart from increasing PON1 activity, lycopene supplementation resulted in a more favorable lipid profile. It improved the concentration of HDL and caused a reduction in elevated levels of total cholesterol, triglycerides, LDL, and VLDL [115].

Lycopene supplementation for 1 month resulted in an increase in PON1 activity in non-diabetic rats [113]. In diabetic rats, lycopene consumption was able to restore PON1 activity, as its basal level was lower in diabetic rats than in a control group. A slight increase in the diabetes–lycopene group and a significant increase in the lycopene group over the control group were found. In another study, the treatment of diabetic rats with lycopene and metformin-induced PON1 activity, an effect similar to that reached by metformin or insulin [116]. The combination of these treatments has a potential beneficial effect of lowering markers of lipid peroxidation, increasing antioxidant defenses, as well as inhibiting postprandial glycemia and dyslipidemia [116]. Thus, lycopene appears as a promising therapeutic agent with the potential to be used in combination therapy to minimize the diabetic complications triggered by glycation and oxidative stress.

#### 4.3.2. The Influence of Lycopene on PON1 Activity and Gene Expression in Clinical Studies

Supplementation of lycopene, as well as the implementation of a lycopene-rich diet to a group of moderately overweight middle-aged subjects, resulted in an increase in PON1 arylesterase activity in serum as well as in HDL2 and HDL3 subfractions [120]. PON1 deficient HDL is dysfunctional and not effective in preventing LDL oxidation. Lycopene may positively affect the structural and functional composition of Apo-AI and thereby restore PON1 activity in HDL particles.

A positive relationship between arylesterase activity and lycopene was also reported in a study on subjects with metabolic syndrome following an energy restriction diet [77]. This association can be explained by the capacity of lycopene to scavenge free oxygen radical products, which would otherwise engage PON1 activity and decrease it [151]. Furthermore, lycopene (and other dietary antioxidants) may exert its effects through modulation of gene expression through regulation of DNA methylation [148]. Methylation of the CpG-rich region overlapping a gene’s promoter is considered a mechanism for inhibiting a gene’s expression [149]. This mechanism was confirmed concerning the PON1 gene, as methylation of the CpG-rich region was found to inversely correlate with PON1 arylesterase activity. Inverse correlations were also observed between methylation of different CpG sites and dietary lycopene, vitamin C, and total tocopherol [77]. At the same time, all measured exogenous antioxidants correlated positively with PON1 arylesterase activity.

In conclusion, lycopene may enhance PON1 expression by inhibiting PON1 gene methylation in subjects with metabolic syndrome. Furthermore, lycopene supplementation was shown to restore PON1 activity in cases of hyperlipidemia, diabetes, obesity, and metabolic syndrome. These changes were found in serum, HDL2, and HDL3. Lycopene may favorably modify the lipid profiles in the population at risk of cardiovascular disease, as well as improve the antioxidant composition of lipoproteins and ameliorate antioxidant defense mechanisms.

### 4.4. The Effect of a Mixture of Carotenoids on PON1 Activity and LDL Oxidation

While it is easier to observe the specific effects of individual carotenoids when supplementing single compounds separately, in nature, they exist in combinations. Therefore, attempts have been made to measure the effect of carotenoids when they are administered together.

Oxidative lipid damage is a marker of the development of cardiovascular disease. PON1 hydrolyses oxidized lipids in LDL, retards atherosclerosis [28], and predicts the development of cardiovascular disease [152]. Similarly, some dietary antioxidants work in this direction, for example, β-carotene protects lipids from oxidation. In vitro and in vivo enrichment of LDL with beta-carotene protected them from cell-mediated oxidation. Surprisingly, this effect was not reached with in vivo treatment with lycopene in this study [153]. However, the administration of lycopene as tomato oleoresin (which contains a mixture of exogenous antioxidants) resulted in a strong inhibition of LDL oxidation. This gives evidence that lycopene may act as an effective antioxidant in synergism with several other natural antioxidants. It is very likely that, when given as a mixture, carotenoids do not only act additively but even synergistically, potentiating each other’s effect [141,154]. This observed effect of higher antioxidant activity of carotenoids, when supplied as mixtures, may be associated with the specific positioning of different carotenoids in membranes [141]. A single oral supplementation of alpha-tocopherol, beta-carotene, lycopene, canthaxanthin, and lutein protected LDL polyunsaturated fatty acids (PUFA) and their cholesterol moieties against oxidative modifications [155]. It has been suggested that the protection from oxidative damage and the associated cardiovascular disease is best achieved by natural antioxidants found in fruit and vegetables.

#### 4.4.1. The Influence of a Mixture of Carotenoids on PON1 Activity in Animal Studies

PON1 activity has been reported to be lower in subjects with type 2 diabetes [156]. Two carotenoids, lycopene and bixin, supplemented individually increased PON1 level and HDL in streptozotocin-induced diabetic rats [117]. The treatment of rabbits on a hypocholesterolemic diet with bixin alone resulted in partial prevention of serum PON1 activity decrease [118]. Adding curcumin to lycopene or bixin led to an even more pronounced effect of decreasing biomarkers of carbohydrate and lipid disturbances, increased HDL levels, decreasing oxidized LDL, and alleviating oxidative stress [117]. Therefore, combining the two antioxidants resulted in a reduction in cardiovascular risk.

#### 4.4.2. The Influence of a Mixture of Carotenoids on PON1 Activity in Clinical Studies

In a randomized controlled trial on subjects with type 2 diabetes, increasing fruit and vegetable intake for 8 weeks resulted in a rise in carotenoids (α-carotene, β-cryptoxanthin, lutein, lycopene) in serum, HDL2, and HDL3, which was accompanied by an increase in PON1 activity in serum and HDL3 [88]. The potential of a high fruit and vegetable diet rich in carotenoids to increase PON1 activity is especially valuable, as PON1 improves the anti-atherogenic mechanisms of HDL and helps to fight the potential negative complications of type 2 diabetes. A positive correlation was found between change in HDL3 β-cryptoxanthin and change in HDL3–PON1 activity, which further supports the idea of using carotenoids to induce the activity of PON1. Yet, this effect is not always observed and depends on the detailed conditions of the study. The consumption of fruits and vegetables for 6 weeks in a group of healthy subjects resulted in a decrease in PON1 activity despite an increase in carotenoids [121]. Another study showed that postprandial PON1 activity raised only after a Mediterranean-like meal together with the increase in carotenoids. On the contrary, the consumption of Western-like meal did not affect postprandial PON1 activity or carotenoids [122]. The effect of dietary modification on carotenoids and PON1 activity in healthy individuals was also assessed by DiMarco et al. While consumption of 2–3 eggs/day increased plasma lutein and zeaxanthin and caused improvements in HDL function, the intake of 3 eggs/day had the additional beneficial effect of inducing PON1 activity [123].

Carotenoids were found to exert their antioxidant effect by the protection of PON1. Yet, the level of PON1 activity preservation by the studied compounds varied in the observed populations. PON1 activity’s correlations with β-carotene, lycopene, lutein, and zeaxanthin were found in Greek but not in Anglo-Celtic subjects [124]. Yet, a different dietary intake of fruits and vegetables, which transferred to a different baseline level of carotenoids, was registered in these two groups, which could influence the outcome. These results also suggest that ethnicity may determine the influence of carotenoids on PON1 activity. Possibly, other factors such as food sources of carotenoids or different preparation methods may affect this relationship. In particular, olive oil usage with vegetables, which was registered to be higher in Greek subjects, may be a confounding factor due to its protective activity towards PON1. Virgin olive oils increased PON1-associated specific activities in a randomized study [157]. Furthermore, after stratification, the observed relationship was significant only in subjects with the R-allele of PON1-192 polymorphism [124].

In other studies, the influence of PON1 gene polymorphism on the modulation of antioxidant activity by dietary antioxidants was also noted. A higher intake of oleic acid was related to an increased PON1 activity only in the PON1-192 RR genotype group [158]. Furthermore, in a study on elderly volunteers, where antioxidant protection offered by components of tomato juice (especially β-carotene and lycopene) was more advantageous in subjects with the R-allele [127]. PON1 activity increased in all volunteers, including the control group. However, antioxidant status improved and LDL-oxidation decreased only in R-allele carriers but not in the QQ genotype group. The same group of authors observed the effect of tomato (as a source of β-carotene and lycopene) and carrot juice (as a source of β-carotene and α-carotene) consumption on PON1 activity and lipid peroxidation in healthy young volunteers for 2 weeks preceded by 2 weeks of low-carotenoid intake. In this setting, as opposed to the previous study, neither of the juices affected PON1 activity. However, tomato juice consumption resulted in a reduction in lipid peroxidation in R-allele carriers in comparison to QQ subjects. Carrot juice did not affect lipid peroxidation regardless of the PON1-192 genotype [127]. Again, the QQ homozygous subjects did not gain any additional antioxidant protection of the lipids with this nutritional intervention. These results suggest that there may be a higher potential for improving the antioxidant defense of PON1 and protection from atherosclerosis through the modulation of HDL function by using dietary antioxidants in subjects with R-allele than in the QQ-genotype. Interestingly, it is the PON1 isoenzyme corresponding to the RR genotype that has a low hydrolyzing activity towards lipid hydroperoxides [159]. Additionally, given that in some populations, subjects of the RR genotype or with R-allele were shown to be at increased risk of coronary artery disease [55,56], it would be indeed very valuable to find ways of enhancement of PON1 activity, particularly in these individuals. PON1 polymorphism modifies the effect of carotenoids on different diseases related to oxidative stress, not necessarily related to atherosclerosis. The distribution of PON1 polymorphism is known to vary between different populations. For the PON1-192 polymorphism, the R-allele was most widely distributed among Mexican (51.7–43.7% depending on the ethnic group) [160], Japanese (65.2%) [161], and Chinese (64.8%) [162] people. Among the Mexican population, the Mestizos have the highest frequency of the RR genotype. Important differences were reported after the comparison of the Mexican and Asian populations to Caucasians. The lowest R-allele frequency was observed in German (22.5%) [163], British (29%) [43], and French (29%) [164,165] populations. Taking into account the allele distribution frequency may help assess the target population, in which carotenoid supplementation improves antioxidant status and limits lipid peroxidation. Another area in which carotenoids may be useful is the osteoporosis risk attenuation. Yet this effect may also be affected by PON1 polymorphism. High serum lycopene was associated with lower bone resorption markers only with subjects with the LL genotype and Q allele [129]. Dietary interventions may be a therapeutic option, applied especially in groups where they offer the greatest advantage. Further research should be encouraged to identify these groups.

In a study on a cohort of 60 Australian Aboriginal people, PON1 arylesterase activity correlated with total carotenoid concentration, as well as the individual carotenoids β-carotene, lycopene, cryptoxanthin, and lutein plus zeaxanthin [125]. In addition, correlations of paraoxonase activity with plasma total carotenoids concentration, due mainly to a strong correlation with lycopene concentration, were found. Dietary and lifestyle intervention in this study increased PON1 activity, homocysteine, and carotenoid concentration. Change in PON1 activity correlated with the change in HDL cholesterol, but the increased HDL cholesterol could not account for all PON1 activity rise. Correlation between carotenoid concentration and PON1 activity were detected at baseline and after the intervention. Yet the authors were not able to find a correlation between change in carotenoids and change in PON1 activity. Not all studies prove that carotenoids influence PON1 activity. Ferre et al., in a study on 388 individuals, found no correlation between β-carotene intake and PON1 activity [126]. The participants of this study were randomly selected with a wide age range (18–75 years) with an equal proportion of men. Kleemola et al. describe an inverse relationship between β-carotene and PON1 activity, but the volunteers in this study were young and healthy university students and employees, mostly women [86]. These conflicting results can be explained, at least in part, by differences in the studied populations.

#### 4.4.3. Conclusion on the Effect of a Mixture of Carotenoids on PON1 Activity

In most of the studies, supplementation of a mixture of carotenoids in their natural form with food increases PON1 activity in serum and/or HDL3. Studies focused on the determination of correlations of individual mixtures of carotenoids with PON1 activity suggest that the relationship between these antioxidants exists only in some populations. While this is observed for most carotenoids and carotenoid mixtures, β-carotene exceptionally shows no correlation or even an inverse correlation with PON1 activity. PON1-192 polymorphism was found to modify the effect of carotenoids on antioxidant status improvement, lipid peroxidation, LDL-oxidation, and bone turnover markers.

## 5. Conclusions

PON1 has been shown to prevent the development of atherosclerosis. Excessive oxidative stress can have a detrimental effect on the enzyme. Evidence exists that carotenoids, among other exogenous antioxidants, possess protective activity over PON1. Their beneficial effects were often shown to depend on the length of time of consumption, the administered dosage, and the type of carotenoid or carotenoids used.

Lycopene is the most effective antioxidant agent among all carotenoids. An assumption can be made that it may also have the most pronounced effect on PON1 activity. Yet, comparative studies in this respect are very scarce. No proof of the superiority of one carotenoid over another was found so far. In a study comparing the effects of lycopene or bixin on PON1 activity, no difference was shown, though both compounds increased the enzyme activity. Another one comparing the effects of tomato juice (source of β-carotene and lycopene) and carrot juice (source of β-carotene and α-carotene) showed no effect on PON1 activity. Comparisons conducted so far rather point to higher effectiveness of carotenoids when they are supplied as mixtures of antioxidants rather than separately.

This is in line with the developing interest in functional foods. Researchers, as well as consumers, are now searching for food, which has value beyond its basic nutritional properties but is rather designed to prevent certain chronic diseases. Carotenoids can also be administered with naturally occurring fruits and vegetables, dietary supplements, or food additives. The research should be oriented toward identifying populations that will benefit the most from their supplementation. So far, intrinsic factors such as PON1 polymorphism, ethnicity, age, and gender were found to determine the effect of carotenoids on enzyme activity.

The review encountered several limitations. There were heterogeneous population characteristics among the studied subjects. The origin, form, or mode of administration of carotenoids may have influenced the outcomes, which was not considered when comparing the effects of the studies and should be a subject of further investigations. Most of the studies were not placebo-controlled. In fact, in case of research concerning fruits and vegetables or juice consumption studies involving placebo are difficult to design. Experiments on larger study groups, with prolonged observations involving randomized controlled trials, are needed to further investigate the role of carotenoids on PON1 activity and gene expression. Studying the interactions between the PON1 gene and its epigenetic regulation may result in finding new possibilities for favorable modifications of the enzyme by therapeutic agents or lifestyle interventions.

Future research on the influence of carotenoids on PON1 activity should develop in two directions. One approach should be based on studying the effects of various carotenoids and their mixtures, forms, doses, and administration methods in the search for those most advantageous, contributing to the highest activity of PON1. Care should be taken to design high-quality, randomized controlled intervention studies on larger study groups, with prolonged observations. The other approach should be more mechanistic. It should aim at explaining the physiological mechanisms, which stimulate PON1 activity by exogenous antioxidants such as carotenoids. The research should aim at understanding the very complex pathways up- and downregulating PON1 gene expression. In addition, the area of epigenetic modification of PON1 by carotenoids should be explored, especially since the first observations are very encouraging. Studying the interactions between the PON1 protein and its epigenetic regulation may result in finding new possibilities for favorable modifications of the enzyme by therapeutic agents or lifestyle interventions.

While many issues regarding the effect of carotenoids on PON1 activity still require further investigation, evidence gathered in this review speaks for the existence of a positive relationship between these antioxidants. Carotenoids supplementation may help modulate PON1 antioxidant and anti-inflammatory effects and increase the enzyme potential to preserve lipoproteins from oxidation and prevent cardiovascular diseases. 

## Figures and Tables

**Figure 1 nutrients-14-02842-f001:**
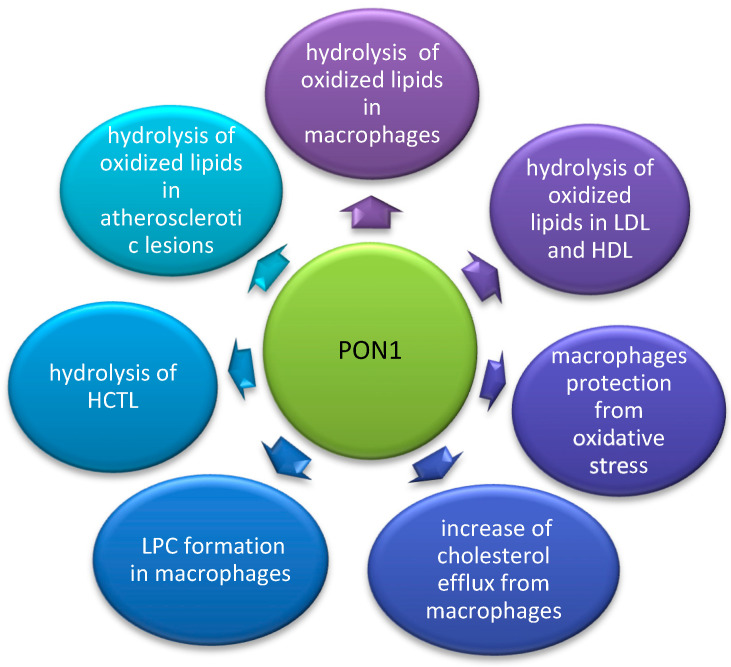
The antioxidative and anti-atherogenic mechanisms of PON1 enzyme. Paraoxonase 1 (PON1), lysophosphatidylcholine (LPC), homocysteine—thiolactone (HCTL).

**Figure 2 nutrients-14-02842-f002:**
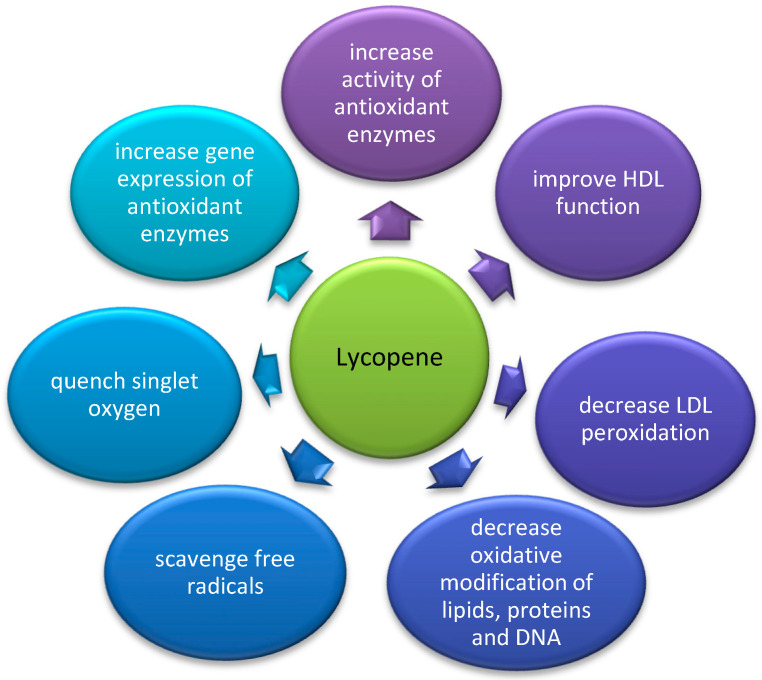
Proposed mechanisms of antioxidant effects of lycopene; (original figure, based on data from [77,120,142,143,144,145,146,147,148,149]).

**Table 1 nutrients-14-02842-t001:** PON1 polymorphisms affecting the enzyme structure and activity.

The PON1 Region		The Affected Site	Effect of the Polymorphism	Ref.
Promoter region	−108C/T polymorphism (rs705379)	The center of consensus binding site for Sp1	Effect on gene expression and serum activity: -Weaker binding of Sp1 in the presence of the T allele than the C allele -Modulation of Sp1 binding affects SREBP2, which upregulates PON1 in the presence of statins	[46,47,48]
	−162A/G polymorphism (rs705381)	The potential NF-1 binding site	Effect on gene expression and serum activity	[46,47]
Coding region	PON1-Q192R (rs662)	Active site	Direct effect on catalytic activity: The 192R allozyme is -more efficient in hydrolyzing paraoxon and chlorpyrifos-oxon, homocysteine thiolactone, higher affinity to HDL binding	[49,50,51,52]
			-less efficient in hydrolyzing diazoxon, sarin, and soman, lower protection against LDL oxidation.	[33,46,49,50,51]
			-no effect on hydrolyzation efficiency of phenylacetate	[49,50]
	PON1-L55M (rs854560) (Possible linkage disequilibrium with the −108 promoter region polymorphism)	The protein structure	Effect on plasma PON1 protein concentration: 55L allozyme has: -higher stability, less susceptible to proteolysis	[50,53]
			-key role in the packing of the protein	[9]
			Effect on PON1 activity: Location of L55M polymorphism in the neighborhood of two crucial amino acids (Glu52 and Asp53), which are required for PON1 activity	[54]

PON1—paraoxonase 1; Sp1—specificity protein 1; SREBP2—steroid regulatory element-binding protein-2; NF-1—nuclear factor-1.

**Table 2 nutrients-14-02842-t002:** Major dietary carotenoids and their common sources.

Chemical Structure	Dietary Source
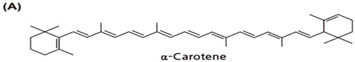	Carrots, squash, pumpkin, palm fruit
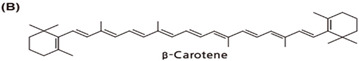	Apricot, carrots, spinach, green collard, cantaloupe, beet, broccoli, tomato, palm fruit, squash, green leafy, mango
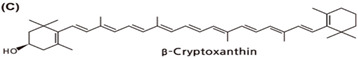	Tangerine, papaya, orange, loquat, tree tomato, persimmon
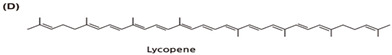	Tomatoes and tomato-based foods (85%), watermelon, pink guava, pink grapefruit, papaya, apricot, Asian gac
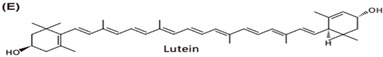	Spinach, green collard, beet, broccoli, green peas, leafy green, corn, corn products, squash, egg yolks
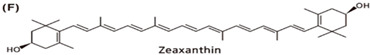	Corn, corn products, squash, egg yolks

**Table 3 nutrients-14-02842-t003:** Overview of animal studies on the effect of carotenoids on PON1 gene expression and PON1 (paraoxonase 1) activities.

Animal Studies				
Study Objective	Study Protocol	Studied Group	Results	Ref.
The effect of asx on PON	Supplementation with 50, 100 and 500 mg/100 g b.w. of asx for 60 days	Hypercholesterolemic rabbits	Restoration of PON by all asx doses	[112]
The effect of asx on PON and ovarian damage	Supplementation with 80 mg/kg b.w. of asx for 14 days	32 female rats in 4 equal groups: control, induced ovarian damage, treated with asx, induced ovarian damage treated with asx	Increase in PON and reduction of ovarian damage	[114]
The effect of lycopene on ARE	Administration of different doses (5, 10 and 50 mg/kg b.w./day) of lycopene for 30 days	Hyperlipidemic rats	Improvement in ARE	[115]
The effect of lycopene on PON	Administration of lycopene for 28 days and comparison of PON between groups	Non-diabetic rats (7 in the control group and 7 in the lycopene group)	Increase in PON	[113]
		STZ-induced diabetic rats (7 in the diabetes group and 7 in the diabetes-lycopene group)	Restoration of PON	
The effects of lycopene or metformin, alone or in combination, on PON	Treatment for 35 days. Assessment of PON in plasma before and after treatment	STZ-induced diabetic rats	Increase in PON	[116]
The effect of treatment with yogurt enriched with lycopene, bixin, lycopene + curcumin, bixin + curcumin on PON	Administration of antioxidants individually or as mixtures for 50 days. Assessment of antioxidants and PON in plasma before, at 10 days, and at 50 days of treatment	STZ-induced diabetic rats	Increase in PON	[117]
The effect of bixin on PON reduced by hypocholesterolemia	60 days of hypercholesterolemic diet alone or with bixin (10, 30, or 100 mg/kg b.w.) or simvastatin (15 mg/kg b.w.) vs. regular chow (control)	42 hypercholesterolemic rabbits divided into 7 groups	Partial prevention of serum PON decrease	[118]

PON—paraoxonase activity; ARE—arylesterase activity; asx—astaxanthin; STZ—streptozotocin.

**Table 4 nutrients-14-02842-t004:** Overview of clinical studies on the effect of carotenoids on PON1 gene expression and PON1 (paraoxonase 1) activities.

Clinical Studies				
Study Objective	Study Protocol	Studied Group	Results	Ref.
The effects of asx on PON1 activities	Collection of blood samples before, 45, and 90 days after supplementation, while regular soccer training.	40 young elite soccer players in two groups (21 asx vs. 19 placebo)	Increase in PON. Interaction effect of asx and training on PON. Increase in PON1 activity towards diazoxon after 90 days in the asx group, and no difference in the placebo group.	[119]
The effect of lycopene on ARE	Treatment with 70 mg lycopene/week. Collection of serum before and after a 12-week intervention	54 moderately overweight middle-aged subjects randomized into 3 groups (lycopene, lycopene-rich diet, and control)	Increase in ARE in serum and HDL_2&3_	[120]
The effect of a lycopene-rich diet (224–350 mg lycopene/week) on ARE				
Assessment of relationships between the ARE with the methylation levels of the PON1 gene transcriptional regulatory region and lycopene	Measurement of ARE and lycopene in plasma, and PON1 transcriptional regulatory region methylation before and after a 6-month energy-restricted dietary weight-loss intervention.	47 obese subjects (46.8% women; 47 ± 10 y.o.; BMI 36.2 ± 3.8 kg/m^2^) with metabolic syndrome	Positive correlation with ARE	[77]
			Increase in PON1 gene expression by inhibition of PON1 gene methylation	
The effects of high and low intakes of vegetables, berries, and apples (containing lutein, β-cryptoxanthin, α-carotene, β-carotene) on PON	Consumption of 1 of 4 controlled isoenergetic diets for 6 weeks containing either 815 or 170 g of vegetables, berries, and apples. Assessment of PON and carotenoids in plasma before and after the diet.	Healthy men and women (*n* = 77; 19–52 y.o.) vs. 19 healthy control subjects	Decrease in PON in all groups; increase in carotenoids in groups on high fruit and vegetable diets in comparison to baseline	[121]
The influence of Mediterranean meal (monounsaturated 61% of fat and antioxidants) vs. Western meal on (saturated 57% of fat) on ARE and carotenoids	Consumption of meals after a 12-h fast, first the Mediterranean meal and after a week of the Western meal. Determination of 0, 2, 4, 7 h postprandial ARE and total carotenoids level in plasma	8 healthy males	Increase in postprandial ARE and total carotenoids only after Mediterranean-like meal	[122]
The impact of consuming 0–3 eggs/d on zeaxanthin, lutein, and ARE	14 wk crossover intervention. Subjects underwent a 2 wk washout (0 eggs/d) followed by sequentially increasing intake of 1, 2, and 3 eggs/d for 4 weeks each. After each period, fasting blood was collected for measurements.	38 healthy men and women (18–30 y.o., BMI 18.5–29.9 kg/m^2^)	Compared with the intake of 0 eggs/d, intake of 2–3 eggs/d promoted a 20–31% increase in plasma lutein and zeaxanthin. Compared with the intake of 1–2 eggs/d, intake of 3 eggs/d resulted in an additional 9–16% increase in serum ARE	[123]
The effect of increased fruit and vegetable consumption on carotenoid content (α-carotene, β-cryptoxanthin, lutein, lycopene) and ARE in subjects with T2D	1- or ≥ 6-portion/day of fruits and vegetable diet for 8 weeks. Collection of fasting serum pre- and post-intervention	80 obese (BMI > 30 kg/m^2^) subjects (40–70 y.o.) with T2D	Increase in ARE in serum and HDL_3_, no change in ARE in HDL_2_	[88]
β-cryptoxanthin correlation with ARE			Positive correlation between change in HDL_3_ β-cryptoxanthin with change in ARE in HDL_3_	
Determination of the relationship of PON and ARE with β-carotene, lycopene, lutein, and zeaxanthin	Measurement of PON and ARE and carotenoids concentration in serum of subjects on habitual diet	127 Greek subjects (men and women; diabetic and non-diabetic equally distributed)	Positive correlation of carotenoids with PON in subjects with the R-allele of PON1–192	[124]
		128 Anglo-Celtic subjects (men and women; diabetic and non-diabetic equally distributed)	No correlation of carotenoids with PON	
Determination of the relationship of total carotenoids with PON and ARE	20 months of diet and exercise intervention. Measurements were taken at baseline and follow-up.	60 Australian Aboriginal subjects (20 men and 40 women; 16–85 y.o.), 38% had T2D	Carotenoids and PON1 activities increased. At baseline: positive correlation with PON and ARE. At follow-up: no correlation of change in PON1 activities with the change of carotenoids.	[125]
Determination of the relationship of individual carotenoids (β-carotene, β-cryptoxanthin lycopene, lutein plus zeaxanthin) with PON and ARE			At baseline: Positive correlation of all individual carotenoids with ARE Positive correlation of lycopene with PON	
Determination of relationship of β-carotene and PON in habitual diet	Assessment of habitual diet by 3-day estimated food record	388 subjects (194 women and 194 men; 18–75 y.o.)	No correlation of β-carotene with PON	[126]
Determination of the relationship of β-carotene and PON in habitual diet	Assessment of habitual diet by 3-day estimated food record	95 healthy young Finnish volunteers (24 male and 71 females)	Inverse correlation of β-carotene with PON	[86]
The effect of tomato juice consumption (rich in β-carotene, and lycopene) on ARE depending on PON1-192 polymorphism	Consumption of 330 mL/day of tomato juice for 8 weeks	50 elderly subjects in 2 groups (control (mineral water) or intervention group (tomato juice))	Antioxidant status improvement and LDL-oxidation decrease only in R-allele carriers. Increase in ARE in intervention group and control.	[127]
The effect of tomato juice consumption (rich in β-carotene, and lycopene) on PON1 activities depending on PON1-192 polymorphism	Consumption of 330 mL/day of juice for 2 weeks after 2 weeks of low-carotenoid intake.	20 young healthy non-smoking subjects were randomized into 2 groups (consuming either tomato juice or carrot juice)	Lipid peroxidation decrease only in R-allele carriers. No effect on PON1 activities	[128]
The effect of carrot juice (rich in β-carotene and α-carotene) on PON1 activities depending on PON1-192 polymorphism			No effect on lipid peroxidation regardless of PON1-192 genotype. No effect on PON1 activities	
Modification of the association between serum concentration of lycopene and oxidative stress markers and bone turnover markers by PON1 polymorphism	Measurement of lycopene, oxidative stress markers, and bone turnover markers in serum	107 women (25–70 y.o.)	PON1 L55M polymorphisms modify the association between lycopene and NTx. The Q192R polymorphism modifies the association between lycopene and BAP. In a subject with RR genotype, lycopene was associated with TBARS.	[129]

PON—paraoxonase activity; ARE—arylesterase activity; asx—astaxanthin; y.o.—years old; b.w.—body weight; BMI—body mass index; T2D—type 2 diabetes; NTx-N-telopeptide of type I collagen, a marker of bone resorption; BAP—bone-specific alkaline phosphate, a marker of bone formation; TBARS—thiobarbituric acid-reactive substances.

## Data Availability

Not applicable.

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
