# Peer review of "Effect of Carotenoids on Paraoxonase-1 Activity and Gene Expression"

_nutrients, 2022, doi:10.3390/nu14142842_

Round 1

Reviewer 1 Report

This review is aimed to summarize current advances of the studies on the effectiveness of the carotenoids on paraoxonase-1 activity and gene expression. The content should be well organized and presented. The manuscript need deep analysis and re-write the summary.

Author Response

Response to Reviewer 1 Comments

Dear Reviewer,

Thank you very much for the revision of manuscript no. 1767061 entitled "Effect of carotenoids on paraoxonase-1 activity and gene expression" authored by Aneta Otocka-Kmiecik. I have now revised the manuscript.

Thank you for your comment that ‘The content should be well organized and presented. The manuscript need deep analysis and re-write the summary”

Response 1:

I have now reanalyzed and reorganized the manuscript, especially the Introduction, Paraoxonase Properties, Paraoxonase Family, PON1 Polymorphism (here a Table is added for clear presentation), The influence of environmental factors on PON1 activity and concentration section, and Conclusions.  I also joined Section 4.1.3 with section 4.1.2. Similarly, section 4.2.2. was merged with section 4.2.3., and 4.3.2 with 4.3.3. to avoid excessive sections.

Response 2:

I have also completely rewritten the summary.

I hope you find these changes satisfactory.

Best regards,

Aneta Otocka-Kmiecik

Reviewer 2 Report

I reviewed the manuscript entitled, Effect of carotenoids on paraoxonase-1 activity and gene expression. The review is well written. Few sections must be improved with details.

Define the Paraoxonase 1 as PON1 for first instance in abstract

Line 10. It is the aim of this review to gather can be changed to Therefore, the aim of this review was to gather……..

Introduction is poorly written. Author (s) must emphasize on paraoxonase-1 activity and gene expression and how carotenoids are showing effect on it. What is the need to conduct this review? Please highlight the research gap and need of this review. Also revise the review objectives

Line 107: after the reference, remove space

Section 4.1. Also include the influence of astaxanthin on gene expression

Section 4.1.3 can be merged with section 4.1.2. Similarly, section 4.2.2. can be merged with section 4.2.1. Follow the same for lycopene.

Also added future recommendations and research need in conclusion

References are not according to journal format. Please revise point-by-point.

Author Response

Response to Reviewer 2 Comments

Dear Reviewer,

Thank you very much for the revision of manuscript no. 1767061 entitled "Effect of carotenoids on paraoxonase-1 activity and gene expression" authored by Aneta Otocka-Kmiecik. I have now revised the manuscript.

Thank you for your comments

Point 1

Define the Paraoxonase 1 as PON1 for first instance in abstract

Response 1:

I have now defined the Paraoxonase 1 as PON1 for first instance in abstract

Point 2

Line 10. It is the aim of this review to gather can be changed to Therefore, the aim of this review was to gather……..

Response 2:

I confirm this change

Point 3

Introduction is poorly written. Author (s) must emphasize on paraoxonase-1 activity and gene expression and how carotenoids are showing effect on it. What is the need to conduct this review? Please highlight the research gap and need of this review. Also revise the review objectives

Response 3

Thank you very much for this comment. I have now rewritten the Introduction and emphasized the effect of carotenoids on PON1 activity and gene expression. I have now tried to emphasize the need for this review and highlighted the research gap and need of this review. I have revised the review objectives.

Point 4

Line 107: after the reference, remove space

Response 4

I confirm this change

Point 5

Section 4.1. Also include the influence of astaxanthin on gene expression

Response 5

Studies on the influence of astaxanthin on gene expression are lacking, which I have now stated in Section 4

Point 6.

Section 4.1.3 can be merged with section 4.1.2. Similarly, section 4.2.2. can be merged with section 4.2.1. Follow the same for lycopene.

Response 6

I confirm merging these sections

Point 7

Also added future recommendations and research need in the conclusion

Response 7

Thank you for this comment. I have now added future recommendations and research need in the conclusion

Point 8

References are not according to journal format. Please revise point-by-point.

Response 8

I have now written the References in Nutrients format.

I hope you find these changes satisfactory.

Best regards,

Aneta Otocka-Kmiecik

Reviewer 3 Report

In this review, Otocka-Kmiecik reported a deepened discussion about the effect of carotenoids on paraoxonase-1 activity and gene expression. In this discussion, the author has gained good attention to the gene structure, polymorphisms, and the anti-atherosclerotic effect of PON1. Moreover, an important aspect treated in this review is the healthy influence of the Mediterranean diet, in particular the carotenoids, on the gene expression and activity of PON1.

The review is well written and easily understandable for the readers. I have only some suggestions for the author:

In my opinion, the review does not adequately address the mechanisms of regulation of the PON1 gene expression, at various levels, in particular at the transcriptional and post-translational levels. These aspects are relevant since they relate to the diet effects on PON1 gene expression.

Line 93, please check the sentence for errors.

Line 111, briefly describe what HCTL is and where it comes from

Line 360, please correct "LDLxidation"

Author Response

Response to Reviewer 3 Comments

Dear Reviewer,

Thank you very much for the revision of the manuscript no. 1767061 entitled "Effect of carotenoids on paraoxonase-1 activity and gene expression" authored by Aneta Otocka-Kmiecik. I have now revised the manuscript.

Thank you for your comments.

Point 1

In my opinion, the review does not adequately address the mechanisms of regulation of the PON1 gene expression, at various levels, in particular at the transcriptional and post-translational levels. These aspects are relevant since they relate to the diet effects on PON1 gene expression.

Response 1

Thank you for this suggestion. The information were added in Introduction section. Also, information on epigenic regulation of the enzyme were added in  "The influence of environmental factors on PON1 activity and concentration" part

Point 2

Line 93, please check the sentence for errors.

Response 2

The sentence was revised and corrected

Point 3

Line 111, briefly describe what HCTL is and where it comes from

Response 3

Thank you for this comment

I have now briefly described what HCTL is and where it comes from

 Point 4

Line 360, please correct "LDLxidation"

Response 4

I have corrected the mistake

I hope you find these changes satisfactory.

Best regards,

Aneta Otocka-Kmiecik

Reviewer 4 Report

Dear author,

manuscript review nutrients-1767061 entiteled "Effect of carotenoids on paraoxonase-1 activity and gene expression" and authored by Aneta Otocka-Kmiecik targets a hot topic that fits well withj the journal scope and that is potentially very interesting to the journal readers. While I appreciated reading this review few issues needs to be adressed before the manuscript meets the journal standards:

1. In the introduction a section should be dedicated to the mechanisms of regulation of PON1 activity (levels of regulation : transcriptional, post-transcriptional, translational or post-translational). Please develop also if there is any epigenetic regulation of the enzyme.

2. In the part dedicated to properties of paraxonase please discuss the 3D structure of the protein. This is important for any post-translational regulation of the enzyme.

3. In the description of paraxonase family please add an alignement of main paraxonase fmembers and highlight the active domains. This make it very easy to follow the manuscript by non specialized readers.

4. For the PON1 polymorphism part plaease make a graphic representation or an alignement to highlight the parts that are important for enzyme structure-function.

5. In the "The influence of environmental factors on PON1 activity and concentration" part please detail if the is any methylation studies or epigenetic regulation of the enzyme.

6. Please argue in part 4 why you selected astaxanthin,  β-carotene and lycopene and what the rational of studying mixtures of these carotenoids.

I am looking towards receiving an improved version of this manuscript that addresses all these issues .

Best regards

Author Response

Response to Reviewer 4 Comments

Dear Reviewer,

Thank you very much for the revision of manuscript no. 1767061 entitled "Effect of carotenoids on paraoxonase-1 activity and gene expression" authored by Aneta Otocka-Kmiecik. I have now revised the manuscript.

I appreciate all your comments. Please, find below the responses to your suggestions.

Point 1

In the introduction a section should be dedicated to the mechanisms of regulation of PON1 activity (levels of regulation : transcriptional, post-transcriptional, translational or post-translational). Please develop also if there is any epigenetic regulation of the enzyme.

Response 1

Thank you for this suggestion. The information on the modulation of PON1 expression was added in the Introduction section as well as information that there is presently very little research on the epigenetic regulation of PON1.

Point 2

 In the part dedicated to properties of paraoxonase please discuss the 3D structure of the protein. This is important for any post-translational regulation of the enzyme.

Response 2

I have now discussed the 3D structure of the protein

Point 3

In the description of paraxonase family please add an alignement of the main paraxonase fmembers and highlight the active domains. This make it very easy to follow the manuscript by non specialized readers.

Response 3

In the description of paraxonase family, I have now added an aligement of main paraoxonase members with a description on which active domains are common for all family members and which are unique for the subfamilies of paraoxonase.

Point 4

For the PON1 polymorphism part plaease make a graphic representation or an alignement to highlight the parts that are important for enzyme structure-function.

Response 4

In the Polymorphism of PON1 section, I have supplied an additional table to show, which parts are important for the enzyme structure-function relationship.

Point 5

In the "The influence of environmental factors on PON1 activity and concentration" part please detail if the is any methylation studies or epigenetic regulation of the enzyme.

Response 5

Thank you for this comment. Information on epigenetic regulation of the enzyme was added in  "The influence of environmental factors on PON1 activity and concentration" part

Point 6

Please argue in part 4 why you selected astaxanthin,  β-carotene and lycopene and what the rational of studying mixtures of these carotenoids.

I have described astaxanthin, β-carotene, and lycopene as the most reviewed literature on the influence of carotenoids on PON1 focuses on these particular carotenoids. I have now supplied this information in part 4. I also explained briefly why these carotenoids are often studied.

Thank you for all your remarks, as I feel that the review has gained a lot from them.

Best regards,

Aneta Otocka-Kmiecik

Round 2

Reviewer 1 Report

This revised review is greatly improved. I have no further comments.

Reviewer 2 Report

Authors are now answered the suggestions made by me. In my opinion, the manuscript can be accepted for publication.